

# A model integrating attention mechanism and generative adversarial network for image style transfer

Miaomiao Fu, Yixing Liu, Rongrong Ma, Binbin Zhang, Linli Wu and Lingli Zhu

School of Information Technology, Luoyang Normal University, Luoyang, China

## ABSTRACT

Image style transfer is an important way to combine different styles and contents to generate new images, which plays an important role in computer vision tasks such as image reconstruction and image texture synthesis. In style transfer tasks, there are often long-distance dependencies between pixels of different styles and contents, and existing neural network-based work cannot handle this problem well. This paper constructs a generation model for style transfer based on the cycle-consistent network and the attention mechanism. The forward and backward learning process of the cycle-consistent mechanism could make the network complete the mismatch conversion between the input and output of the image. The attention mechanism enhances the model's ability to perceive the long-distance dependencies between pixels in process of learning feature representation from the target content and the target styles, and at the same time suppresses the style feature information of the non-target area. Finally, a large number of experiments were carried out in the monet2photo dataset, and the results show that the misjudgment rate of Amazon Mechanical Turk (AMT) perceptual studies achieves 45%, which verified that the cycle-consistent network model with attention mechanism has certain advantages in image style transfer.

# INTRODUCTION

In recent years, the research on image style transfer develops rapidly, and various forms of image style conversion emerge endlessly. Due to the vigorous development of Internet short video, image style transfer is known by more and more people. Various special effects on social media platforms are favored by the vast number of netizens, which also makes the application of image style transfer more extensive. The evolution of image art styles has been a long-standing development, progressing from non-realistic imagery to the current era of image style transfer. From the initial rendering of artistic images to the present applications like Prisma and Ostagram, which are popular worldwide and rely on image transfer technology, this journey has underscored the growing attention given to research in image style transfer.

Corresponding author
Miaomiao Fu, fmm@lynu.edu.cn

For the image style transfer, academia and industry have proposed different technical methods. *Gatys, Ecker & Bethge (2015b)* for the first time, reproduced the image style of famous paintings on natural images through convolutional neural network (CNN). First, they pre-trained the content of the photos through CNN, modeled the features, and further counted the style characteristics of the famous paintings. Then, CNN was used to extract the photo content and the style of the famous painting, and the style was matched with the target photo, and the stylized image with specific artistic characteristics was successfully generated for the first time. The "pix2pix" framework proposed by *Isola et al. (2017)* learns the mapping from input image to output image through conditional generative adversarial network (GAN), which is the first method of image style migration using generative adversarial network in a real sense, but it requires paired matched data. Both methods need to be trained with supervised information, and it is very difficult to obtain high-quality labeled data sets. At the same time, traditional convolutional networks pay much attention to features in the image neighborhood while giving up global features during training, and lack the ability to deal with long-distance dependence. However, the attention mechanism could abstract image features into two feature spaces, and then multiply the input of the attention layer with proportional parameters, and finally add them to the input image data, this operation enhances the feature representation of the target content information, and suppresses the feature representation of non-target content.

Based on unsupervised learning cycle-consistency network (*Liu, Breuel & Kautz, 2017*; *Dong et al., 2017*; *Huang et al., 2018*) is an image-to-image conversion method, the core idea is to design the inverse mapping from the target domain to the initial domain, and introduce cycle-consistency loss to force the initial domain to generate the target domain result while generating the initial domain result, so that this result is as similar as possible to the original initial domain to constrain the target domain (*Zhu et al., 2017*). In summary, this paper proposes a generative adversarial network model that integrates attention mechanism and cycle-consistency network, and introduces the attention mechanism in the generator part of the network (*Goodfellow et al., 2014*; *Liu & Tuzel, 2016*; *Pan et al., 2017*). In the generator part of the network, the attention mechanism abstracts image features into two feature spaces, then multiplies the input ratio parameter of the attention layer, and finally adds back to the input image data. This method can prioritize the weights in the neighborhood and then assign more weights to the non-neighborhood features, so the introduction of the attention mechanism enables the generator to pay attention to all the information of the feature matrix when generating images and improve the imaging quality.

## RELATED WORKS

Style transfer is to obtain the style characteristics of a certain type of atlas, and add this style feature on the basis of retaining the content characteristics of the target image to realize the transformation from one style to another. In the early days of the image style work, the researchers focused on image texture transfer, because in a sense, the local statistic of texture can be used as an abstract description of the painting style. However, it is limited

to express the texture of the entire image only through the low-level features of the image, and it is impossible to ideally express the semantic recognition ability that style transfer should have, and it is impossible to effectively transfer the style according to the semantic content. The previous image style transfer method consists of two parts: texture modeling and image reconstruction, in which texture modeling is divided into two categories: (a) parametric texture modeling method based on statistical distribution (b) non-parametric texture modeling method based on MRF (Markov Random Field); Image reconstruction is divided into two categories: (a) slow image reconstruction methods based on online image optimization (b) fast image reconstruction methods based on offline model optimization.

Gatys, Ecker & Bethge (2015a), the pioneer of image style work, used a parametric, slow style transfer algorithm based on statistical distributions in neural style. Gatys first proposed a texture modeling method based on Gram matrix in NIPS, the core of which is a recurrent neural network-based method (Gatys, Ecker & Bethge, 2015b). The style is modeled with the Gram matrix, the weight of the Gram matrix is updated by the gradient descent method by the slow image reconstruction method, and the updated pixel value is restored when the image is reconstructed, so that the feature matrix of the original image and the style map are as similar as possible, and the final image can not only save the content map texture, but also inherit the style of the style map. Li & Wand (2016a) proposed a non-parametric texture modeling method based on MRF, which differs from traditional MRF in that it adopts a similar scheme to the Laplace neural style transfer method proposed by Li et al. (2017a), which can well retain local structure and other information in the image.

Image reconstruction is mainly divided into two categories: online and offline, among which the offline model can use the pre-trained model to solve the problem of large training volume and slow speed. The single-mode single-style algorithm is the earliest fast stylization algorithm, and its core idea is to train a model for each style, and when an image needs to be generated, a content map is input into the forward model, and a result output map based on a specific style can be obtained in the output. In industrial production, only need to package and upload this forward model, and the reserved input and output interface can quickly obtain stylized migration images in use.

The fast stylized transfer algorithm of PSPM (Per-Style-Per-Model) mainly includes (1) parametric fast stylized single model single style algorithm based on statistical distribution, this algorithm mainly has two representatives, Johnson, Alahi & Fei-Fei (2016) and Ulyanov et al. (2016). Both use a forward network calculation style and a style. The loss function is the same as the slow stylization algorithm employed by Gatys, which is stylized using the Gram matrix statistic. However, the specific network architecture of the two is different, He et al. (2015) using residual network design, Li et al. (2018) using multi-scale network. (2) Non-parametric single-model single-style algorithm based on MRF. This method was first proposed by Li & Wand (2016b) as a Markov generative adversarial network method, which accelerates the slow stylization algorithm in Li & Wand (2016a). The discriminant model in the GAN network is used to replace the previous patch matching and train the forward network. Although the results are not very ideal, its theoretical ideas have great value and are one of the methods discussed in this article. Later, this method was further developed in Isola et al. (2017).

The fast stylized transfer algorithm of single model and multiple styles has a larger market in industrial applications, and each style of single-style method needs to be trained and uploaded separately, which is very detrimental to a large number of style development. Multi-style in the multi-style model is not "real multi-style", but a multi-style based on the same type of atlas, for the same type of style, each atlas to train a single-model single-style network is very wasteful, in this case, the Google Brain team developed a basic idea of MSPM-based algorithm (*Dumoulin, Shlens & Kudlur, 2016*). This method can unearth the shared parts between different style networks, and then change only the different parts for the new style, and the shared parts remain unchanged. In this way, only a few parameters in the CIN (Conditional Instance Normalization) need to be changed to get the new style result. The "style banking" approach proposed by *Chen et al. (2017)* is similar to *Dumoulin, Shlens & Kudlur (2016)*. After solving the problem of how to use a network to store multiple styles, you need to consider how to use the desired style. *Zhang & Dana (2018)* of Amazon AI proposed a multi-style generator method, the core idea is to combine the style features extracted from the VGG network with the middle-layer feature map in multiple scales in the stylized network through the Isnpiration Layer, that is, add a layer of VGG network as an input signal in front of the network, and then decide which stylized image to generate through the input signal. *Li et al. (2017b)* uses pixels as input signals in a feed-forward network-based method, first binding the style map to the noise map, and then concatting the feature map of the middle layer as a signal to decide which style to adopt.

Single-model arbitrary style rapid transfer algorithm, arbitrary style model (ASPM) expresses a zero-shot-fast style transfer idea, new style does not need to be trained. The earliest ASPM algorithms were the patch-based algorithm proposed by *Chen & Schmidt (2016)*. The core idea of the algorithm is to use the CNN feature space to match and exchange the content and style patch, and finally obtain the combined feature weight, and use the fast image reconstruction algorithm to reconstruct the exchanged feature. This is similar to MRF's non-parametric ASPM algorithm, but it is not real-time and takes more time. *Huang & Belongie (2017)* built a method that could do ASPM in real time. Inspired by CIN, he proposed Adaptive Instance Normalization. AdaIN uses a data-driven approach to normalize content directly into different styles by training on large-scale styles and content. Google Brain's arbitrary image style transfer method (*Ghiasi et al., 2017*) is also through a data-driven ASPM approach. It can be seen as a follow-up work of *Bahdanau, Cho & Bengio (2014)*, which gives a different style by changing the affine transformation parameters in the CIN. In this way, Google Brain trained a style prediction network specifically to predict affine transformation parameters for each style. However, the data-driven approach makes stylized renderings correlate with the variety and number of styles in the training set. The style learning-free approach of *Li et al. (2017c)* uses a series of feature transformations for ASPM, avoiding the limitations of data-driven. ZCA whitening transform can erase the style information in a picture, retain the complete high-level semantic information, and use coloring transform to migrate colors. *Wei et al. (2022)* put forward a comprehensive overview of existing text-to-image synthesis techniques, which mainly include the text encoding, text-direct image synthesis, and text-guided image synthesis.

The main objective of discussion in this paper is the rapid stylization algorithm for single mode and single style. The core idea is to train a model for each style, and when you need to generate an image, enter a content map into the forward model, and you can get a result output map based on a specific style in the output. In industrial production, only the forward model needs to be packaged and uploaded to the reserved input and output interface, and the stylized migration image can be quickly obtained in use.

## MODEL DESIGN

In order to make the generated style transfer images completer and more accurate, this paper uses the effect of attention mechanism on target feature acquisition and non-target feature inhibition, combined with the forward and backward feedback learning principles in cycle-consistency networks, to construct a new generative adversarial network model for style transfer. Models are divided into generative networks and discriminant networks. Among them, the generative network is composed of the forward network and attention network of the cycle-consistency network, and the discriminant network is composed of the reverse network of the cycle-consistency network.

The forward network is responsible for information acquisition of the input image, gradually learning and extracting the target style and content contained in the input image, while the attention network is responsible for the model to enhance and extract the possible feature information of the target content and style at the key stage, and suppress the feature information of the non-target content and network. The discriminant network is responsible for restoring and reconstructing the encoded feature information as indiscriminately as possible.

### The forward network of a circular consistency network

In the process of graph-to-graph conversion, in order to solve the problem that pictures are not paired, a cycle-consistency adversarial network is designed, and the generator is composed of encoder, converter and decoder in the generation network (as shown in Fig. 1). The encoder completes the extraction of input image features during the down-sampling process, and uses a two-layer convolutional neural network to encode the image from $64 \times 64$ feature vectors to $256 \times 256$ feature vectors to ensure that the model can obtain more image details and ensure that the gradient oscillation is reduced during multiple iterations, which is conducive to the convergence of the model. While ensuring that the depth and parameters of the generator network remain unchanged, a layer of attention layer is inserted before the generator converter to help the model learn from the corresponding feature vector containing the target content and style feature information. The output of attention and the output of the encoder are identical in shape, but this feature information is optimized by attention. After receiving the image feature vector encoded by the encoder, the channel attention mechanism module performs an average pooling operation on the feature map, and then passes through two layers of perceptron, using ReLU and Sigmoid as the activation layers respectively, and finally generates the attention feature map. The converter uses the trained style features to retain the content in the content map, transfer the style to the target domain, and obtain the feature vector of the target style, the converter

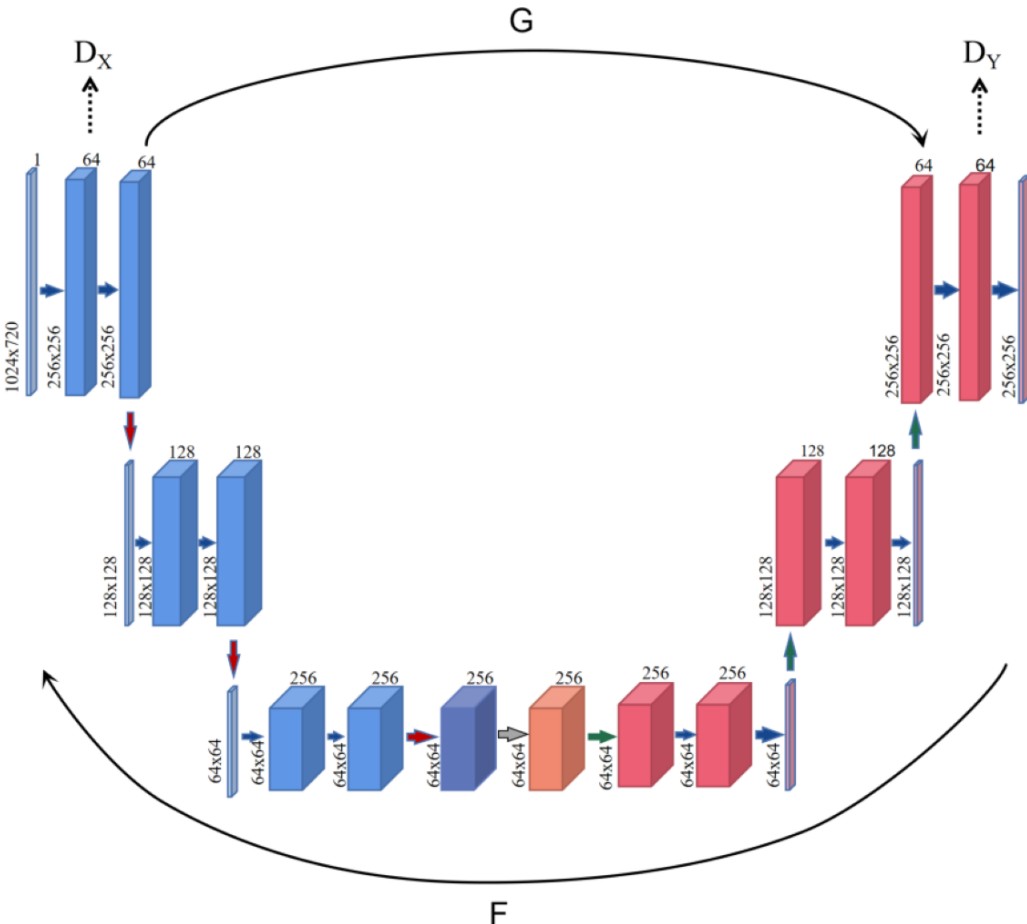

**Figure 1** The overview architecture of proposed model.

follows the residual network idea in the fast style transfer, and uses six ResNet modules, each module is a neural network composed of two convolutional layers, which can achieve the purpose of retaining the original image features at the same time during conversion. The decoder reconstructs the image during the up-sampling process, and uses deconvolution to restore low-level features from the feature vector to ensure the integrity of the reconstructed image.

## Attention network

In the process of cycle-consistency network from the source domain X to the target domain Y, first the sample set x is input into the generator G, and then the generator G generates y' according to the coding feature information to make the discriminator $D_y$ misjudge, and the discriminator $D_y$ determines whether y' belongs to the target domain Y also makes the discriminator $D_y$ trained. The specific operation is shown in Fig. 2.

At the same time, in order to strengthen the association relationship between long-distance target features in feature information, this paper uses the attention mechanism to enhance the association relationship between pixels of the encoded feature information

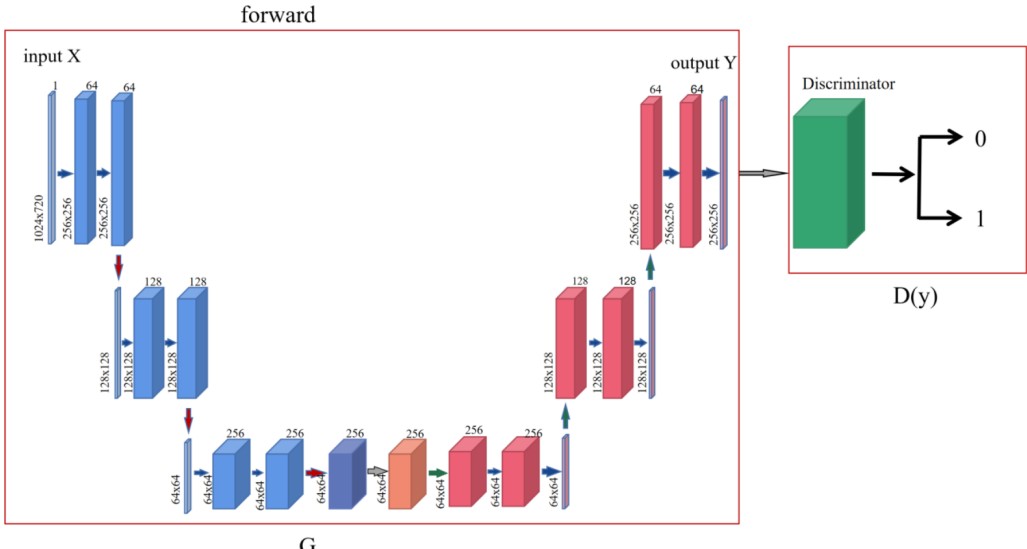

**Figure 2** **Forward network structure diagram of cycle-consistency network.**

obtained from the forward network from the circular consistency network, and the specific operation is shown in Fig. 3.

In Fig. 3, the Q, K, and V vectors represent the query, index, and value, respectively, and the Q vector and K vector are normalized by Softmax, and the final result is multiplied by points with the V vector to obtain the feature weight map to ensure the normal operation of the converter.

Attention can abstract it into two stages, the first stage calculates the weight coefficient based on query and key, and the second stage weights value based on the weight coefficient. The first stage can be abstracted into two stages according to the function, the first stage calculates the similarity and correlation between the two according to query and key, and the second stage normalizes the correlation matrix. The final calculation process is shown in Fig. 4.

In stage 1, the author matches the weights of Q vector and K vector according to the vector dot product of the two and obtains them, and optimizes the weights according to their index allocation.

$$\text{Sinilarity}\left(\text{Query}_i, \text{Key}_i\right) = \text{Query}_i \cdot \text{Key}_i \tag{1}$$

In stage 2, the author introduces a calculation method similar to SoftMax to convert the result value of the first stage, on the one hand, it can be normalized, and the original calculated score can be sorted into all element weights and a probability distribution of 1. On the other hand, the weight of important elements is more prominent through the internal mechanism of SoftMax.

$$a_i = \text{Softmax}\left(\text{Sim}_i\right) = \frac{e^{\text{Sim}_i}}{\sum_{j=1}^{L_x} e^{\text{Sim}_i}} \tag{2}$$
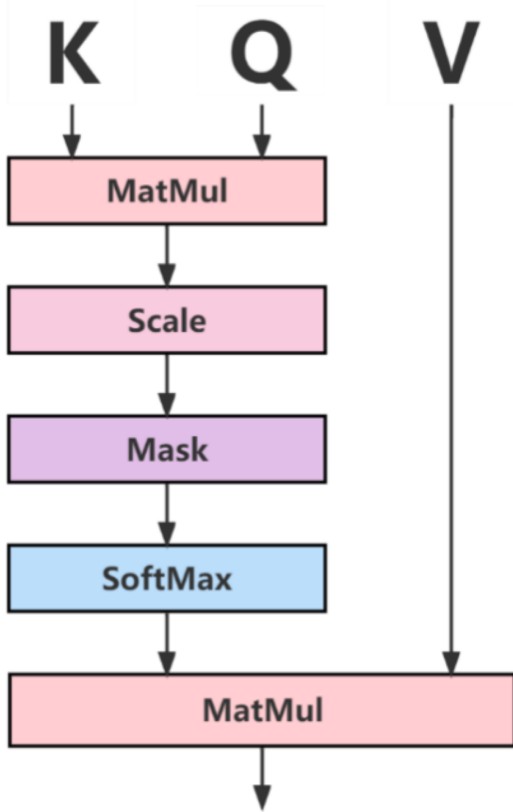

**Figure 3** **Attention mechanism diagram.**

In this stage, the attention mechanism solves the long-distance dependence problem, which can distribute the normalized results to all positions, and continuously optimizes the three vectors in the backward process to further solve the problem of convolutional neural networks in GAN networks with difficulty in long-distance dependence.

$$\text{Attention}\left(\text{Query}, \text{Source}\right) = \sum_i = 1^{L_x} a_i \cdot \text{Value}_i \tag{3}$$

### Circular consistency network reverse network

After the input image X is processed by the generation process of the cycle-consistency network and the feature enhancement processing of the attention module, the reverse process of the cycle- consistency network is introduced in order to make the image content y′generated by the model X learn an accurate mapping distribution. Among them, the discriminant network adopts the PatchGAN discriminant model proposed by *Isola et al. (2017)*, and PatchGAN proposes to divide the image into several blocks, classify the authenticity of the image in chunks, and finally calculate the average of the results, and draw the classification conclusion according to the comprehensive score. The approximate process is shown in Fig. 5.

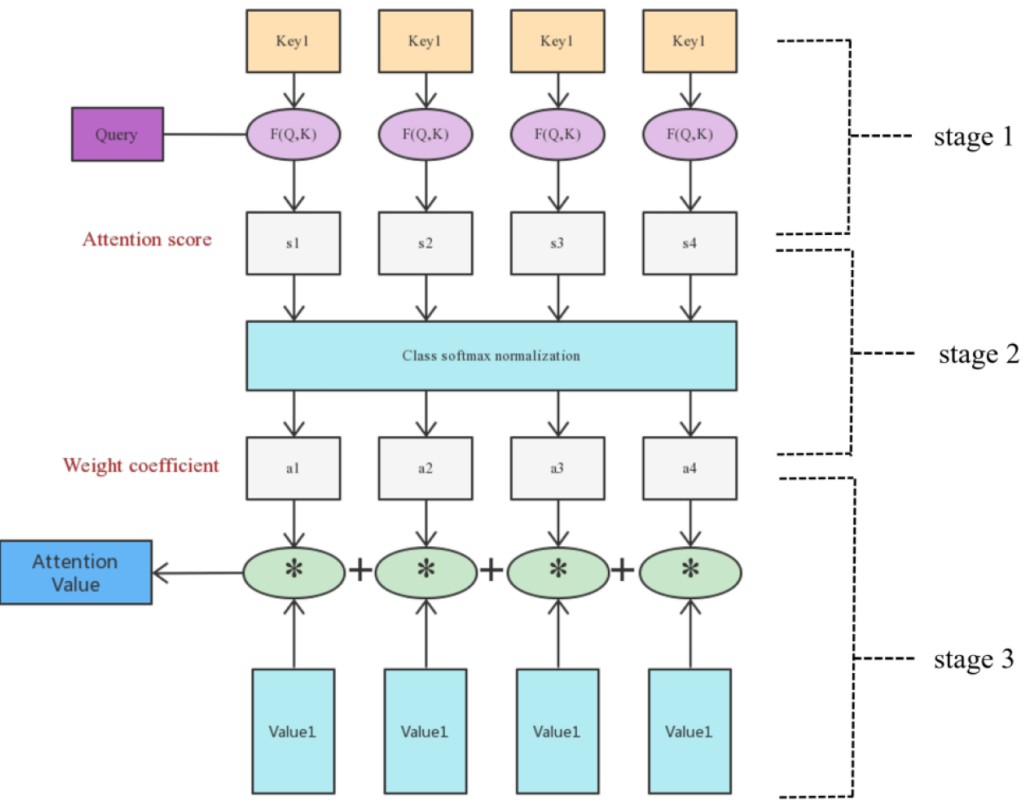

**Figure 4** Attention mechanism calculation process diagram.

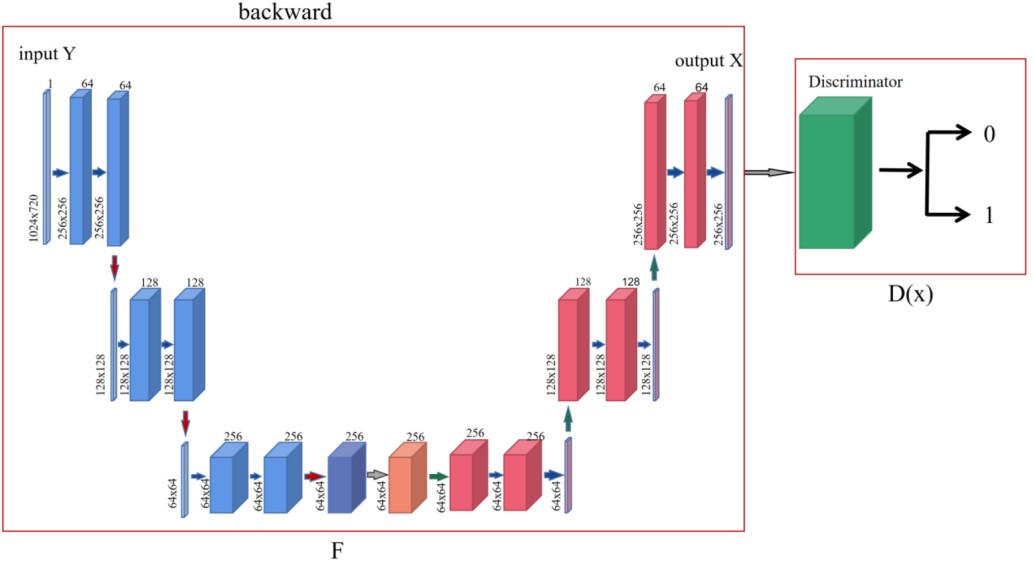

**Figure 5** Cycle-consistent network discriminant network diagram.

**Table 1  The parameter configuration of the platform in our experiments.**

| Hardware and software names | Parameter |
|---|---|
| Intel CPU | 128G DDR4 |
| SSD | 2TB |
| Nvidia GeForce RTX 3090 GPU | 24G |
| Ubuntu | 18.04.6 |
| Python | 3.8 |
| torch | 1.10.2 |
| CUDA | 11.3 |

**Table 2  The detail parameter settings when training a model.**

| Item | Paramter |
|---|---|
| n_epochs | 200 |
| dataset_name | monet2photo |
| batch_size | 8 |
| lr | 0.0002 |
| b1 | 0.5 |
| b2 | 0.99 |
| n_cpu | 2 |
| channels | 3 |
| sample_interval | 100 |
| checkpoint_interval | -1 |
| n_residual_blocks | 9 |
| lambda_cyc | 10.0 |
| lambda_id | 5,0 |
| img_height_width | 256*256 |
| Attention_ channels | 64 |

# EXPERIMENT AND RESULT ANALYSIS

## Experimental parameters and datasets

This experimental platform is a programming environment for the deep learning framework Pytorch built on the Ubuntu 18.04.6 system, and the software and hardware environment settings used are shown in Table 1. Table 2 displays the detail parameters when training a model on our platform.

The experimental dataset uses the monet2photo dataset provided by CycleGAN, which was jointly released by Jun-Yan Zhu, Taesung Park, Phillip Isola, Alexei A. Efros on March 30, 2017, monet refers to Monet-style paintings, and photo refers to the images taken, which is used to transfer the styles of the two images. At the same time, it consists of 1,337 Monet-style oil paintings of $256\times256$ size to form the X-domain training set, and 3,671 photos of $256\times256$ size to form the Y-domain training set. Figure 6 shows a display of the Monet2photo dataset.

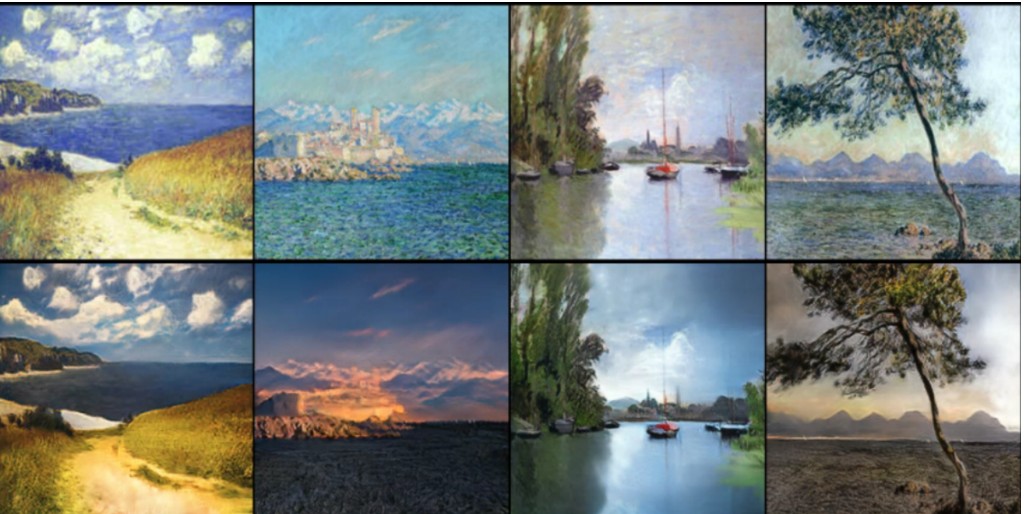

**Figure 6** **Monet2photo dataset display.** Source: Monet2photo dataset, CC-BY-SA-NC 4.0.

## Evaluation metrics

Previous studies have shown that the design and selection of evaluation indicators have always been a difficult problem for unsupervised tasks because they do not have accurate labels to help discriminate experimental results. Many unsupervised learning tasks will choose FCN-score for evaluation tasks. The core idea is to use fully convolutional neural network (FCN) (*Long, Shelhamer & Darrell, 2015*) to learn the source domain of a large number of original pictures, so that the FCN network remembers the real image features as much as possible, and then the real picture and the generated picture are mixed into the FCN network. Finally, the score is based on the judgment results. Due to the unreliable stability of the algorithm, this paper does not adopt this method of "artificial intelligence evaluation artificial intelligence". This paper adopts the method of AMT perceptual studies (*Zhang, Isola & Efros, 2016*), randomly mixes real pictures and generated pictures into a group of five interviewed students, conducts eight sets of experiments for each student, and finally takes 40 sets of valid results, each group of pictures is displayed for 1 s, let them select real pictures and real oil painting pictures, and finally comprehensively evaluate their scores. For each algorithm tested, there were 25 experimenters, and each experimenter viewed multiple pairs of experimental images, each containing a real or false picture, which could be a map or photo, and clicked on the picture that he thought was the real image. The first 10 times are used for familiarization, and its selection will have corresponding feedback, telling whether it is correct or not, and then the next 40 experimental data will be used as the basis for scoring. In the original CycleGAN paper, an approach called AMT perceptual studies was employed. In the 'map aerial photo' task, the author conducted real and fake cognitive studies on Amazon Mechanical Turk (AMT) to determine the simulation of the output of the author's development.

Table 3 shows the results of AMT evaluation of the two types of networks. According to the 40 rounds of experiments designed by the author in each group, the introduction of

**Table 3   Evaluation results of AMT for two types of networks.**

| Experimental group | Misjudgment rate |
| --- | --- |
| CycleGAN oil painting turned into photo | 30% |
| MyGAN oil painting turned into photo | 30% |
| CycleGAN photo turned into oil painting | 40% |
| MyGAN photo turned into oil painting | 45% |

**Table 4   Loss scoring results for two types of networks.**

| Name | Loss value |
| --- | --- |
| CycleGAN discriminant network | 0.090076 |
| MyGAN discriminant network | 0.242549 |
| CycleGAN generative network | 1.532771 |
| MyGAN generative network | 0.697493 |

the attention mechanism strengthens the details of style transformation to a certain extent, making the generated photos more realistic, and reducing the recognition between the generated oil paintings and the real oil paintings. Moreover, the probability of misjudgment of the generated oil painting is 45%, which is much higher than that of the photo generated without the attention mechanism (*Kong et al., 2024*), where the false positive rate refers to the probability that the obvious difference between the real photo and the generated photo cannot be distinguished.

The loss value in Table 4 is the mean error of the two generators or two discriminators, using the calculation method of the minimum absolute error, as can be seen from Table 4, the final average loss of the two method generators and discriminators reaches a very small state, which is very close to the required Nash equilibrium state.

Therefore, the experimental results show that adding the attention mechanism in the model coding stage can effectively make the model learn more about the relationship between colors, strengthen the rationality of the generated results, and make the generated images more delicate or realistic.

## Loss function

At the same time, the average absolute value error (MAE, $L_1$ Loss) (*Zhu et al., 2017*) is used in this paper, which refers to the error obtained by the absolute value between the predicted value F(x) and the true value y, and *n* represents the number of experiments.

$$\text{loss}(x, y) = \frac{1}{n} \sum_{i=1}^{n} |y_i - f(x_i)| \tag{4}$$

The detailed expressions in this article are:

$$l(G, F) = \mathbb{E}_{x \sim P_{data}(x)} \left[ \| F(G(x)) \|_1 \right] + \mathbb{E}_{y \sim P_{data}(y)} \left[ \| G(F(y)) \|_1 \right] \tag{5}$$

where $\| F(G(x)) \|_1$ represents the error obtained by discriminator $F$ on the absolute value between the predicted value of $G(x)$ generated by sample $x$ through the generator and the

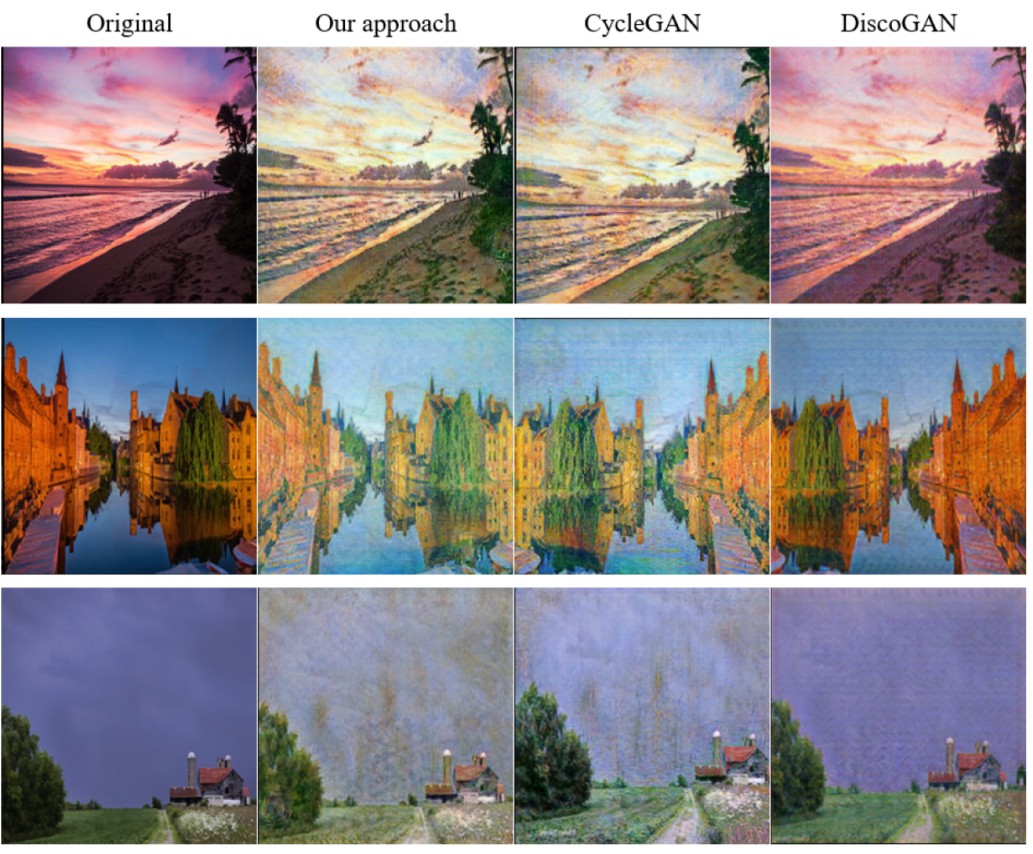

**Figure 7 Monet-style oil painting turned landscape.** Source: Monet2photo dataset, CC-BY-SA-NC 4.0.

true value y, and $\| G(F(y)) \|_1$ represents the error obtained by discriminator $G$ on the absolute value between the predicted value of $F(y)$ generated by sample $y$ by the generator and the true value $x$.

## Experimental results

The results of two sets of experiments based on the monet2photo dataset are shown in Figs. 7 and 8. The first column is the input images, the second column corresponds the generated image of our model, the third column is from the CycleGAN model, and the fourth column is the ones of the DiscoGAN network. Through these data, the effect of images generated by different GAN networks can be reflected, and then the role of their network models can be analyzed.

The experiments show that how to transform Monet's oil paintings into images of real life as much as possible. As can be seen from the result plots of Fig. 7, the image details generated by the DiscoGAN (*Kim et al., 2017*) network are not well generated, such as the chaotic deformation of the trunk texture in the third column of images, which is far from the effect of the other two networks in the experiment. The grassy part of the first image generated by the CycleGAN network is too dim, resembling a burnt brown, and the distant sea surface is cloudy and does not reflect the color of the sky. Comparing with the results of

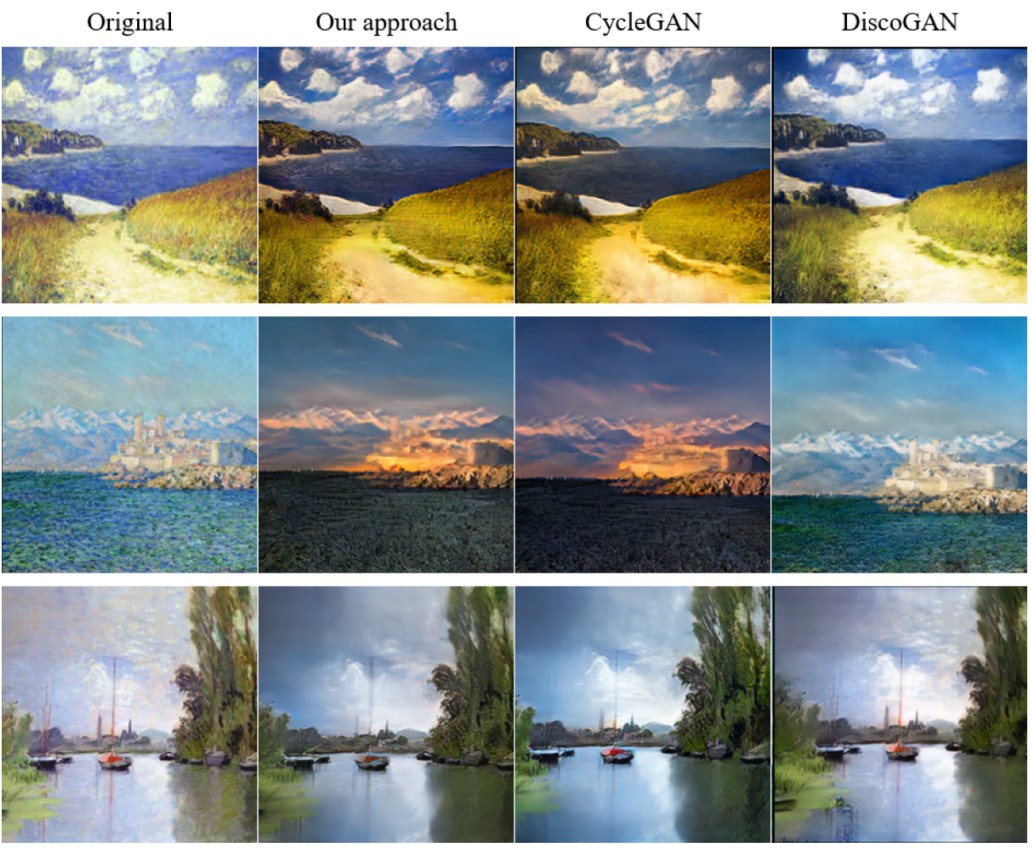

**Figure 8** **Landscape to Monet-style oil painting.** Source: Monet2photo dataset, CC-BY-SA-NC 4.0.

other SOTA models, the ones generated by the proposed model, the excessive grass color is more natural, the color is more realistic, and the sea surface has obvious ripple. In the third, the color of the water surface generated by CycleGAN is too vivid, which does not conform to the effect of dark weather generation, and the presented method generates the water surface to learn the dark color of the sky, and the generation effect is more realistic. The above conclusion shows that the proposed method achieves better results in the task of converting Monet's oil painting to real image. This is because the attention mechanism in the coding stage helps our method obtain more feature information of the input images, then the model can better pay attention to the color relationship between the grass color and the nearby dark area in the image, the color relationship between the water surface and the sky, etc. The above content makes the image generation is more natural and more in line with the color and light and shadow in the overall environment in the image.

The experiments in Fig. 7 is to convert real photos into oil painting images with a more detailed brush stroke. From the image in the first row, it can be seen that compared with the CycleGAN generated image, the image generated by our proposed model has a higher degree of restoration to the original photo and better than detail processing. As can be seen in the third line of images, the color grasp of the image generated by the method in this paper is more in place, which is more in line with the vivid texture of the color

in the monet2photo image set. The above conclusion shows that the proposed method achieves better results in the task of converting real photos to Monet-style oil painting, because after adding the attention mechanism in this model, the model can pay attention to more information between pixels in the image, so that the color and brush strokes of the generated oil painting image are more delicate.

The above comparison results show that the proposed model can achieve the effect of practical application, and a large number of experiments verify that the attention mechanism in the coding stage utilizes its advantages in associating the context of long-distance dependence to accelerate the learning of the model, improve the image generation effect, and reduce the model training batch. The results are more natural and the colors are more vivid, so that the model can better notice the color relationship between the grass color in the image and the nearby dark area, that the work make the image learning more natural. It can effectively make the model learn more about the relationship between colors, strengthen the rationality of the generated results, and make the generated images more delicate or realistic. The desired effect is achieved, which can provide some help to GAN network learners in learning feature information.

## CONCLUSION

To address the long-distance dependencies between pixels of different styles and contents, we built a novel model by fusing an attention module and residual blocks. The attention module eliminates some useless feature information and concentrates the available resources of the computer during the learning process to help the proposed model learn the styles and content features of the image. The residual module alleviates the gradient issue that may arise during model training. Finally, experimental results on two datasets indicate that the presented model constructed in the paper has achieved competitive results.

### Funding

This work was supported by the Science and Technology Innovation Team of Henan University (No. 22IRTSTHN016), special project of key research and development plan of Henan Province under Grant no. 221111111700, the teaching reform research and practice project of higher education in Henan Province in 2021 (2021SJGLX502), the Key Science and Technology Program of Henan Province (No. 222102110366), the Science and Technology Research Project in Henan Province (No. 232102320068), the Key scientific research projects of colleges and universities in Henan Province (No. 24A520026). There was no additional external funding received for this study. The funders had no role in study design, data collection and analysis, decision to publish, or preparation of the manuscript.

### Grant Disclosures

The following grant information was disclosed by the authors:
The Science and Technology Innovation Team of Henan University: No. 22IRTSTHN016.

Special project of key research and development plan of Henan Province: 221111111700.
The teaching reform research and practice project of higher education in Henan Province in 2021: 2021SJGLX502.
Key Science and Technology Program of Henan Province: No. 222102110366.
Science and Technology Research Project in Henan Province: No. 232102320068.
The Key scientific research projects of colleges and universities in Henan Province: No. 24A520026.

## Competing Interests

The authors declare there are no competing interests.

## Author Contributions

- Miaomiao Fu conceived and designed the experiments, analyzed the data, performed the computation work, prepared figures and/or tables, authored or reviewed drafts of the article, and approved the final draft.
- Yixing Liu conceived and designed the experiments, performed the experiments, analyzed the data, performed the computation work, prepared figures and/or tables, authored or reviewed drafts of the article, and approved the final draft.
- Rongrong Ma conceived and designed the experiments, performed the experiments, analyzed the data, performed the computation work, authored or reviewed drafts of the article, and approved the final draft.
- Binbin Zhang conceived and designed the experiments, authored or reviewed drafts of the article, and approved the final draft.
- Linli Wu conceived and designed the experiments, authored or reviewed drafts of the article, and approved the final draft.
- Lingli Zhu conceived and designed the experiments, authored or reviewed drafts of the article, and approved the final draft.

## Data Availability

The Monet2photo Dataset is available at https://tianchi.aliyun.com/dataset/93932. This data was available at the time of review/acceptance.

The code is available at GitHub and Zenodo:

- https://github.com/star0718/mygan_Image_Style_Transfer
- star0718. (2024). star0718/projectproject: First release of my awesome software (V1.0.0). Zenodo. https://doi.org/10.5281/zenodo.10815826.

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
