# Peer review of "A model integrating attention mechanism and generative adversarial network for image style transfer"

_PeerJ Computer Science, doi:10.7717/peerj-cs.2332_

## Round 0.1 · original submission · Major Revisions

Please revise the article carefully according to the comments. Then it will be evaluated again.

Reviewer 1 ·

Basic reporting

In summary, this article constructs a generative model for style transfer using a recurrent consistency network as the backbone framework and integrating attention mechanisms. At the same time, the ResNet module is used in the converter module to improve the stability of the model, avoid gradient vanishing, and make the training process more stable.
(1) The experimental results show that the model can effectively transfer image styles and has a certain degree of innovation.
(2) The format of the period punctuation in line 89 is incorrect. Please make the necessary corrections.
(3) Please reorganize the long sentence into several shorter ones to make it easier to read. Including the following:
Lines 31 to 36: "Gatys [1] et al., for the first time, reproduced the image style of famous paintings on natural images through convolutional neural network (CNN), they first pre-trained the content of photos through CNN to perform feature modeling, and further made statistics on the style characteristics of famous paintings, and used CNN to extract the content of photos and the style of famous paintings to match the style to the target photo, then the stylized images with given artistic features are successfully generated for the first time."
Lines 227 to 232: "Previous studies have shown that because they do not have accurate labels to help the identification of experimental results, the design and selection of evaluation indicators has always been a problem, many unsupervised learning tasks will choose FCN-score for evaluation tasks, the core idea is to use fully convolutional neural network (FCN) [34] to learn its source domain for a large number of original pictures, so that the FCN network remembers the real picture features as much as possible, and then the real picture and the generated picture are mixed into the FCN network."
(4) For formulas, there are some inconsistent symbols before and after. Please check if the formulas in the entire text are correct and if the symbols are consistent throughout the text.
(5) There are some language and grammar errors, the authors should go through the paper and check the errors. Please re-organize the long sentences, a very long sentence should be splitted to a couple of shorter ones to make it more readable.
(6) For the references, there are some are not consistent, also for some references,the format is not suitable for the paper, please check it and make it more clearly.

Experimental design

no comment

Validity of the findings

no comment

Additional comments

no comment

·

Basic reporting

The English needs to be more conspicuous than lengthy.
Raw data and source code are both shared.
The field background is sufficient for readers to gain field knowledge.
Overall, this article is okay in terms of technology and organization. The writing needs to be improved.

Experimental design

Existing deep models fail to solve the long-distance dependence between pixels in terms of different image styles and contents. Aimed at this problem, the authors propose an image-style transfer model that integrates attention mechanisms and residual modules. In order to learn the typical style and content features, an attention mechanism is introduced to eliminate useless features and concentrate the available resources. Furthermore, with the aim of making the training process more stable, a residual module is designed to alleviate the gradient problem during the model training process. Extensive experiments are conducted to evaluate the proposed model, and the results show that it can effectively improve the image generation effect.
However, there are still some problems that the author needs to solve.
(1) The attention mechanism shows great results for the style feature transfer. Please explain it in detail.
(2) The residual module alleviates the gradient problem during model training. Please explain it in detail.
(3) To achieve greater clarity, please break up lengthy sentences throughout the whole text and restructure some of the complex ones for readability.
(4) Some paragraphs are formatted incorrectly. Please proofread the whole text.
(5) Some sentences are grammatically inappropriate. Please carefully check the whole text.

Validity of the findings

The novelty needs to be highlighted as a separate section somewhere in the text.
Conclusions need to be improved with some supporting results.

---

## Round 0.2 · accepted · Accept

Thanks to the authors for their efforts to improve the work. This version successfully satisfied the reviewers. I can be accepted currently. Congrats!

Reviewer 1 ·

Basic reporting

This paper is throughly revised and thus can be accepeted.

Experimental design

no comment

Validity of the findings

no comment

Additional comments

no comment

·

Basic reporting

no comment

Experimental design

no comment

Validity of the findings

no comment

Additional comments

The authors have addressed my questions and this article can be accepted.